# Older or Wiser? Age and Experience Trends in 20 Years of Olympic and World Swimming Championships Open Water 10-km Races

**DOI:** 10.3390/jfmk6040089

**Published:** 2021-10-29

**Authors:** Luis Rodríguez-Adalia, Santiago Veiga, Jesús Santos del Cerro, José M. González-Ravé

**Affiliations:** 1Catalonian Swimming Federation, Diputació St., 237, 08007 Barcelona, Spain; dtnatacio@natacio.cat; 2Faculty of Sports Sciences, Universidad de Castilla-La Mancha, Carlos III Avenue, 45008 Toledo, Spain; 3Health and Human Performance Department, Universidad Politécnica de Madrid, Martin Fierro St., 28024 Madrid, Spain; 4Department of Statistics, Universidad de Castilla-La Mancha, 45008 Toledo, Spain; jesus.scerro@uclm.es; 5Sports Training Laboratory, Faculty of Sports Sciences, Universidad de Castilla-La Mancha, Carlos III Avenue, 45008 Toledo, Spain; josemaria.gonzalez@uclm.es

**Keywords:** age of peak performance, swimming, performance analysis, experience, talent development

## Abstract

The aims of the present research were to estimate the age of peak performance (APP) and to examine the role of previous experience at the world-level open water race performances. Finishing positions and age of swimmers (639 females and 738 males) in the 10-km events of World Championship (WCH) and Olympic Games (OG) from 2000 to 2019 were obtained from the official results websites. Years of previous experience were computed using the number of previous participations in WCH or OG. APP was estimated using quadratic models of the 10th percentile top race positions and resulted in 28.94 years old for males (R^2^ = 0.551) and 27.40 years old for females (R^2^ = 0.613). Regression analysis revealed an improvement of 1.36 or 8.19 finishing positions for each additional year of age or experience, respectively (R^2^ = 0.157). However, significant differences (*p* < 0.001) between age and experience showed that the swimmer’s age became less relevant for performance as years of experience increased. These results, in terms of age, are in line with other mass-start disciplines of similar duration (≈2 h) and, in terms of experience, confirm the importance of previous participation in improving tactical decision making during open water races.

## 1. Introduction

Open water swimming is an endurance discipline that became part of the Olympic programme in 2008, with events ranging from one to five hours duration [1]. The most common race distances for open water events are 5 km, 10 km, and 25 km, although only the 10-km race is held during Olympic Games (OG) [2]. This means that most elite athletes prioritise their preparation for this event in comparison to the 5-km or 25-km distances [3].

Until now, the age of the successful open water competitors during Olympic races (2008, 2012, and 2016 editions) has been situated in a wide range from 24.1 to 28.5 years old for men and from 19.8 and 26.9 for females (www.olympic.org (accessed on 20 September 2020)). One previous study estimated the age of peak performance (APP) in open water races at 23.4 ± 0.9 years old for females and 25.6 ± 1.6 for males [4]. This estimate was made by selecting the annual ten fastest times in world-level races from 2000 to 2012. However, swimming speeds were compared from races where external factors, such as water temperature, currents, race circuit, and even the athletes’ strategies, were probably different [2]. The APP has also been calculated in different endurance disciplines such as running [5,6], cycling [7,8] and triathlon [9], as it can be of paramount importance in the correct planning of long-term athlete development programmes [10]. In 2015, Allen et al. [10] determined a relationship between race duration and APP in a variety of sport disciplines, indicating that APP ranged from 20 years-old for swimming pool events (lasting two minutes) to 39 years-old for ultra-distance cycling (lasting 27–29 h). In pool-swimming races, it has been reported that APP decreased with increasing event distance [6,11,12] and that women achieve APP around two years before male swimmers [11]. Interestingly, pool-swimming races presented the youngest APP estimation compared to other endurance disciplines, including open water events.

Open water is a head-to-head discipline with a mass-start format where all athletes are competing in the same lane and rewards are based on finishing position instead of finishing time [13]. Compared to swimming-pool events, open water races present environmental conditions such as water temperature, tides and currents that are often changing and are out of control the competitors’ control [14]. These two aspects highlight the great tactical component of open water events, where competitors usually avoid a time-trial pacing profile (as in swimming-pool events), as they must be able to adapt to the external race stimuli [15]. Recent studies suggest that pacing strategies and mid-race positioning highly influence 10-km race results [1,3,13]. In line with the high level of hydrodynamic drag, swimmers in these events actively aim to swim behind or at the side of a surrounding competitor—the drafting effect [16]—as a means of maintaining an optimal race situation with the lowest energy cost [17]. Indeed, successful competitors at the world stage usually employ a conservative strategy with rear mid-race positioning and an end-spurt in the last quarter of the race distance [1,3,13,18].

In this context, the accumulation of competitive experience for open water swimmers can be a cornerstone of performance, as it can enable them to properly make decisions and respond adequately to the race stimuli [17,19]. Open water swimming specialists are usually fast swimmers in pool events, and it is common for some athletes to compete in both the open water and pool races within the same international competition [20]. In 2019, Gregorio Paltrinieri, the world champion in 1500 m swimming-pool events, competed in the 10-km for the first time in a World Championship (WCH) and declared post-race that “*I cannot understand why I’m not at top level yet. (...) They (open water top level swimmers) are more experienced, they know what to do in every race situation*” [21]. By repeatedly competing in open water events, athletes can optimise some performance determinants not directly dependant on the athlete’s physiology, but more related to correct race strategy decision-making, mid-race group positioning, nutritional strategies, or even mental resilience [10]. Indeed, the estimated APP in mass-start endurance disciplines such as marathon running, Ironman triathlon, and ultra-endurance running events tends to be greater than for head-to-head disciplines in separated lanes such as track running or pool-swimming [10]. This may be linked to the higher level of experience of athletes with a higher age of participation (defined as years of high-level competition), as previously in swimming-pool events [22].

Currently, there is no previous research examining the effect of previous experience or swimmers’ age on performance during open water races. Therefore, the aims of the present research were: (1) to estimate the APP in open water races; and (2) to examine the role of previous experience at the world level on open water race performances. It was hypothesised that APP would be greater for open water races than for pool-swimming disciplines and that greater experience in open water would be related to better race performance.

## 2. Materials and Methods

### 2.1. Participants

An observational retrospective study was conducted in accordance with the Helsinki declaration. A database was developed using historical data from the Omega (http://www.omegatiming.com (accessed on 5 May 2020)) and FINA (http://www.fina.org/ (accessed on 5 May 2020)) official results websites. Final time, final position, and participants’ age were retrieved from all 10-km events from WCH and OG from 2000 to 2019. This comprised three editions of the OG (2008, 2012, and 2016) and sixteen editions of the WCH. It should be noted that the first edition of the open water World Open Water Swimming Championships was held in 2000, after which it was held every year until 2011 and every two years thereafter. The final database contained 639 records of female and 738 records of male swimmers, although swimmers who were disqualified or did not finish were excluded from further analysis. The local University Ethical Committee approved this research on 30 November 2016. No informed consent was necessary, as the data are based on publicly available resources.

### 2.2. Data Analysis

The research design included the swimmer’s age and years of experience in open water major events as explanatory variables of race performances. Swimmers’ experience was computed as the first, second, third, etc. participations of each swimmer in the OG or WCH 10-km open water races. Experience data was calculated since 2005 to avoid bias in the first four editions of WCH: in first edition all participants had one year of experience, in the second edition all participants had one or two years of experience, etc. For interpretation purposes, the top-10 finishing positions in each 10-km race were considered to be successful participants, following criteria used in previous research [13]. This metric was used as the top-10 positions in the WCH race the year prior to the OG are directly classified for the next OG edition.

### 2.3. Statistical Analysis

Mean values, ratios, coefficients of dispersion, and double-decker graphics were calculated for the specified explanatory and performance variables to provide the basic characteristics of the swimmers under analysis. A general linear model (GLM) was used to examine the association between the dependent variable (finishing position in the 10-km OG and WCH races analysed) and the independent variables (age and experience). The R^2^ and global significance test coefficients were calculated to verify this model. A *p*-value of less than 5% in the linear regression model was considered significant. Model assumptions of normality and homoscedasticity were assessed using the Kolmogorov–Smirnov test and residuals graphics, respectively, showing a satisfactory pattern for all residuals. Analysis of covariance (ANCOVA) was used to determine the interaction between experience and final position, using age as the covariate.

Additionally, APP was analysed using quadratic models of the 10th percentile of top positions in both male and female categories. The statistical model (parabolic or quadratic regression) and theoretical background according to Anderson [8] was employed for this purpose. Statistical analyses were conducted using R software (v. 3.6.1 for Windows).

## 3. Results

The age of elite open water swimmers competing in the 10-km races of the WCH and OG since 2000 are displayed in Figure 1. The age of winners in the male races ranged from 17.6 to 33.4 years old, whereas age of winners in the female races was between 17.8 and 32.2 years old. Age values for the mean top-10 finishing positions were both above or below the age of winners or top-3 swimmers, depending on the edition. The years of experience for the participants in the 10-km races of major competitions is presented in Table 1, in conjunction with the average finishing positions they achieved. According to the data, each accumulated year of experience (from one to four) represented an improved average finishing position in both genders.

Table 2 shows the regression analyses for the finishing positions in the 10-km races of the WCH and OG based on age and experience as explicative variables. The model indicates an acceptable degree of goodness in the adjustment (R^2^ = 0.157), with a significant association (*p* < 0.001) observed between age and experience. According to β estimates, the analysis revealed that one more year of age in a swimmer improves their race results by 1.36 positions. In addition, one more year of experience in competition represents an improvement of 8.19 positions in the race result. Analysis of covariance (ANCOVA) also showed significant differences (*p* < 0.001) between age and experience in competition. The graphical representation of the model (Figure 2) illustrated the relationship between years of experience and age of swimmers. It suggested that greater experience indicates a better position and, also, that the age of the swimmer is less relevant when the swimmer had participated in more WCH and OG.

Finally, the APP was estimated using quadratic models of the best swimmers (10th percentile top race positions), showing significant *p*-values (*p* < 0.001) for all coefficients in the quadratic function. Male swimmers presented an APP of 28.94 years old, whereas female swimmers achieved estimated peak results at 27.40 years old (Figure 3). In both models (males and females), the quadratic coefficient was positive, indicating that the parabola reached a minimum and the values of the coefficients (10th percentile in each of the swimmers’ ages) were higher before and after the peak. In both models, the R^2^ coefficients indicated an acceptable degree of goodness of fit, around 60% for both the male (R^2^ = 0.551) and female (R^2^ = 0.613) models.

## 4. Discussion

The present study aimed to estimate APP in open water races and to examine the role of previous experience at the world level in open water race performances. Data compiled from 20 years of WCH and OG 10-km races revealed that greater age and years of experience increased the chances of achieving top-10 finishing positions, although age was a poorer predictor if the number of previous participations was greater. Compared to swimming-pool disciplines, APP in open water races was considerably greater and confirmed the importance of learning how to manage the changing race conditions.

Top-10 competitors in the 10-km open water races of the WCH and OG since 2000 ranged from 22.4 to 28.1 years old in the male races and from 22.4 to 27.8 in the female races. Winners and medallists in the female races tended to be younger from 2006, when the International Olympic Committee announced that open water disciplines were becoming part of Olympic programme. However, general age trends seemed to be stable across the 20 years of analysed competitions (Figure 1). APP was estimated by quadratic regression, with a result of 27.4 and 28.9 years old for males and females, respectively (Figure 3). These data are in line with most physical capacities peaking around the age of 30 years old and with physical attributes such as the aerobic capacity or the movement economy generally increasing with training experience and thus age [10]. There is only one previous study in the open water discipline [4], which estimated APP to be considerably lower for males (25.6) and females (23.4) than in the present research. However, Zingg et al. [4] based their calculations on the fastest times per year in 10-km international level races, and it is now well acknowledged that time or velocity comparison can be misleading due to the changing external conditions such as water temperature, currents, and circuit structure between open water races [2]. This is the reason why, in the present study, quadratic regression of APP was performed according to the finishing positions and not swimming velocities. The 1.5-year difference in APP between male and female swimmers should also be noted. This was in line with findings in other endurance sports [5,9] and swimming-pool events [11], and could be explained by the earlier onset (≈2 years) of puberty in females compared with males [11].

Compared to other disciplines, the APP in open water races was similar to other mass-start competitions of similar duration (≈2 h), such as marathon (28.8 years old for males and 29.8 for females in winners or top-5 positions during world-level races) [5] or Olympic-distance triathlon (27.6 years old for males and 27.1 for females in the top-16 of World Cup or OG races) [9]. However, APP in open water was considerably greater than that estimated for top-16 competitors during swimming-pool distance events in OG (21.9 years old for females and 22.9 for males) [11]. These results are in line with observations by Baldasarre et al. [2], who indicated different age trends in swimming-pool versus open water races and could be explained by the large tactical component of mass-start events and the beneficial effect of drafting behind an opponent [17]. The high hydrodynamic water resistance means that open water swimmers group themselves and try to swim right behind (or at the side) of the leading competitor (drafting effect), aiming for a decrease in the energetic cost of locomotion [16]. This could be the main reason that the pacing decisions of open water swimmers according to their group positioning or packing behaviour seem to be critical for the final race results [1,3,13].

In the open water discipline, competitions usually present a slow development of the race as athletes tend to adapt to the specific race dynamics and, consequently, select the pacing strategy that allows them to obtain the best finishing position [17]. Specifically, successful participants in 10-km races have been reported to generally present a conservative pacing strategy with mid to rear group positioning in the first half of the race, maximum time-gaps with race leaders of 15–20 s, and an important end-spurt in the last race lap [1,3,13]. These characteristics suggest that the accumulation of competitive experience through the years could help open water swimmers to successfully cope with tactical race demands [10,17,19] and to deal with changing environmental conditions, improve their own mental resilience, or develop specific nutritional strategies [10]. Indeed, this could explain the observed greater APPs in open water compared to swimming-pool events, where most of the external conditions are controlled and the characteristics of the pacing profile are similar to time-trial sports [15].

The results of the present research confirmed the hypothesis that repeated participation over the years in the OG or WCH open water races results in an estimated improvement of 8.19 positions for each year of participation (Table 2). This is a greater effect than previously reported in swimming-pool WCH participations [22] and it is probably connected to the increased experience of open water athletes, allowing them to better anticipate the actions of their opponents and to adequately respond to the relevant cues from the race environment and from their own internal state [17,19]. More specifically, regression analysis illustrated (Table 1) that four or more years of previous participations could be needed to optimise open water race performances. These data are in line with the approximately four years required to reach peak racing speed in highly tactical events such as ultra-endurance running [23]. However, in the present research, we also observed a statistical interaction between age and years of experience, indicating that the importance of age in open water results decreased if the previous experience was greater (Table 2). Participants probably needed a minimum number of previous experiences on the open water stage to be competitive, but this could be achieved earlier if participation in open water events began at younger ages. Indeed, considering that APPs in swimming-pool events fall between 21 and 22 years old, and that successful open water competitors present higher speed in middle- and long-distance pool swimming than unsuccessful competitors [20], the accumulation of experience through previous participations may be able to anticipate APP in open water races. Of course, one of the limitations of the present research is that the previous experience was not evaluated in events other than the OG or WCH. Successful participants in open water races could have obtained the necessary experience in other types of events, such as the World Cup or Continental Championships. However, pacing behaviour in running mass-start events has been observed to be dramatically adjusted in important events such as the OG and WCH [17], which may suggest that it is necessary to analyse these races differently.

## 5. Practical Applications

Knowledge of the specific APP of open water events could provide coaches and sports organizations with valuable information to design long-term training plans. The present study revealed important differences in the age of elite open water versus pool-swimming athletes, despite both disciplines usually working in similar training or team structures. Beyond the age of competitors, the number of previous participations in major open water competitions seems to represent a key determinant of overall race success. A minimum of four years of participations in the major open water events seem to be necessary to optimise the overall race results on the world stage. Future research should examine the developmental path of youth athletes who progress into the elite level in the open water discipline. In this way, the career performance trajectories of the open water athletes could be designed.

## 6. Conclusions

The present research reported, for the first time, the APP and previous experience of competitors in 10-km open water races. According to results of the OG and WCH since 2000, successful competitors presented their peak performances at 27.40 and 28.94 years old (females and males, respectively), and experienced at least four prior participations in these world-level competitions. Up to the APP, the older the swimmers became and the more participations they accumulated represented an improvement in their finishing positions, although age was a poorer predictor if the number of previous participations was greater. Due to the tactical predominance of a mass-start discipline such as open water races, participants probably need a minimum number of previous experiences on the world stage in order to be competitive, and this can be achieved earlier if participation in open water events begins at younger ages. These results are of critical importance for the design and implementation of long-term training plans for open water swimmers.

## Figures and Tables

**Figure 1 jfmk-06-00089-f001:**
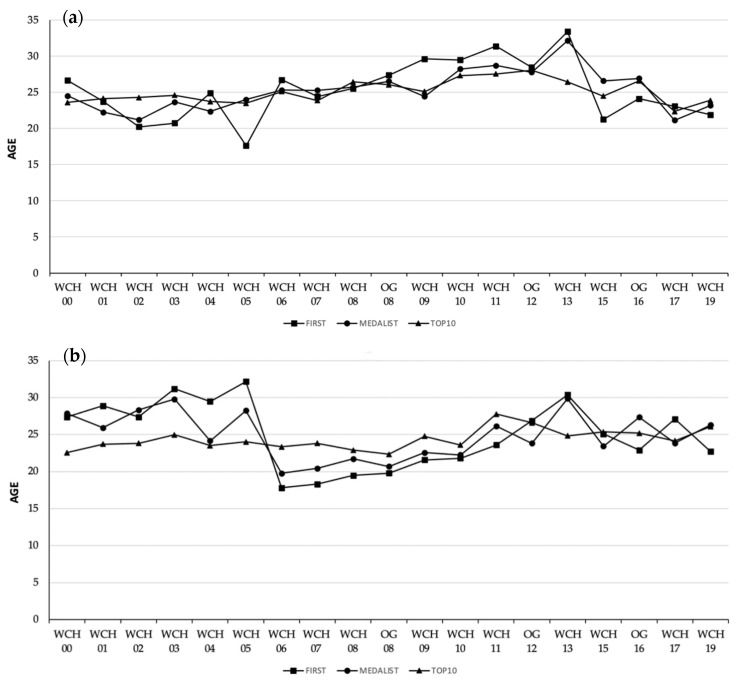
Age of successful (**a**) men and (**b**) women participants in the World Swimming Championships (WCH) or Olympic Games (OG) 10-km open water races from 2000 to 2019. Note: “WCH 00” refers to the 2000 edition, “WCH 01” to the 2001 edition, etc.

**Figure 2 jfmk-06-00089-f002:**
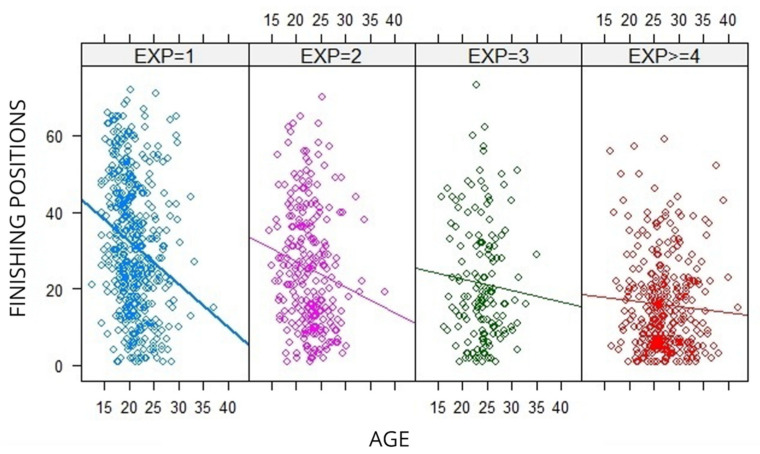
Regression analyses for finishing positions in the 10-km races of the WCH and OG since 2000, based on age and experience of swimmers. The four modules from left to right represent swimmers with one, two, three, and four or more years of experience.

**Figure 3 jfmk-06-00089-f003:**
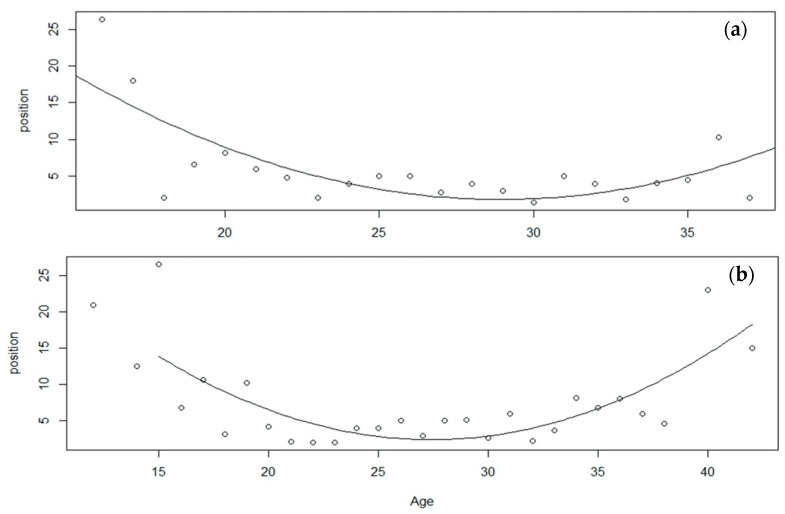
Quadratic models of the top 10th percentile of men (**a**) and women (**b**) participants in the 10-km races of the WCH and OG since 2000.

**Table 1 jfmk-06-00089-t001:** Years of accumulated experience (from one to seven) and average finishing positions of swimmers in the 10-km races of the WCH and OG since 2000.

Years of Experience	Female		Male	
	*n*	Mean Finishing Position ± SD	*n*	Mean Finishing Position ± SD
1	211	27.93 ± 15.63	252	34.10 ± 18.26
2	116	24.34 ± 13.61	135	25.99 ± 17.26
3	74	19.00 ± 13.68	80	23.69 ± 16.74
4	53	16.79 ± 13.64	52	15.52 ± 11.04
5	36	17.31 ± 14.25	38	16.29 ± 10.52
6	22	13.32 ± 9.62	22	16.59 ± 11.27
7	15	18.47 ± 12.74	15	16.13 ± 15.07

**Table 2 jfmk-06-00089-t002:** Regression analyses for finishing positions in the 10-km races of the WCH and OG since 2000, based on age and experience of swimmers.

	Estimate (β)	Std. Error	t Value	*p*-Value	R^2^
(Intercept)	62.29	3.65	17.05	*p* < 0.000	0.157
Age	−1.36	0.16	−8.54	*p* < 0.000	
Experience	−8.19	0.02	−4.80	*p* < 0.000	
Age: Experience	0.16	0.04	3.61	*p* < 0.000	

## Data Availability

The data that support the findings of this study are available from the corresponding and first authors (santiago.veiga@upm.es and dtnatacio@natacio.cat) upon reasonable request.

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
