# Peer review of "Older or Wiser? Age and Experience Trends in 20 Years of Olympic and World Swimming Championships Open Water 10-km Races"

_jfmk, 2021, doi:10.3390/jfmk6040089_

Round 1
Reviewer 1 Report
jfmk-1430231
Lines 31-32: add a reference
Lines 32-34: add a reference
Lines 34-35: add a reference
Lines 36-38: add a reference
Lines 48-49: for which disciplines?
Line 50: add a reference
Line 55-57: add a reference
Lines 70-71: add a reference
Line 80: Ironman
Lines 81-83: add a reference
Line 100: how many years for the sixteen editions?
Figure 1: add the unit at the y-axis and the x-axis is nearly impossible to read
Line 267: what are the practical applications, the implications for future research?
Author Response
Response to Reviewer 1 Comments
First of all, we would like to thank the reviewers for their detailed revision of the manuscript and for their valuable suggestions that surely will improve the quality of the manuscript. Each of the reviewer comments have been addressed below and changes have been indicated on the manuscript with red colour. Additionally, a small correction was done in lines 41 and 213 about the APP results of previous studies
Point 1: Lines 31-32: add a reference
Response 1: Added
Point 2: Lines 32-34: add a reference
Response 2: Added
Point 3: Lines 34-35: add a reference
Response 3: Added
Point 4: Lines 36-38: add a reference
Response 4: Added
Point 5: for which disciplines?
Response 5: Disciplines were specified as requested by reviewer.
Point 6: Lines 50: add a reference
Response 6: Added
Point 7: Lines 55-57: add a reference
Response 7: Added
Point 8: Lines 70-71: add a reference
Response 8: Added
Point 9: Ironman
Response 9: Modified as requested
Point 10: Lines 81-83: add a reference
Response 10: Added
Point 11: Line 100: how many years for the sixteen editions?
Response 11: The years for the WCH events are specified in the following sentence: “It should be noted that the first edition of the open water World Open Water Swimming Championships was held in 2000, after which it was held every year until 2011 and every two years thereafter”
Point 12: Figure 1: add the unit at the y-axis and the x-axis is nearly impossible to read
Response 12: Following the reviewers recommendations, some changes has been done in Figure 1 and 3.
Point 13: Line 267: what are the practical applications, the implications for future research?
Response 13: It has been included in Practical applications section
Reviewer 2 Report
ABSTRACT
Well presented and structured
INTRODUCTION
The introduction provides enough background information for readers to understand the research aim, however, the authors should clarify the importance of this topic.
I would recommend summarizing the introduction to clarify the topic and provide a concise research aim
METHODS
The methodology proposed looks appropriate, well designed, and conducted.
RESULTS
Results include the most relevant data.
All of the tables explain in a correct direction the data obtained.
The quality of figures are poor
DISCUSSION
Discuss the importance and novelty of the data obtained
The conclusion should concisely respond to the research aim
Explain applications according to the study conclusion
English language must be check
Author Response
Response to Reviewer 2 Comments
First of all, we would like to thank the reviewers for their detailed revision of the manuscript and for their valuable suggestions that surely will improve the quality of the manuscript. Each of the reviewer comments have been addressed below and changes have been indicated on the manuscript with red colour. Additionally, a small correction was done in lines 41 and 213 about the APP results of previous studies
Point 1: The quality of figures are poor
Response 1: Following the reviewers recommendations, some changes has been done in Figure 1 and 3.
Reviewer 3 Report
Interesting article in which the authors decided to estimate the age of peak performance in open water races and to examine the role of previous experience at the world level on open water race performances.
In the introduction, based on the literature, the state of knowledge on research on competitors taking part in long-distance swimming competitions is presented.
The methods are well described, allowing you to recreate all the analyzes.
The results are presented in tables and graphically. Based on them, we can better understand the relationship between age of peak performance and previous experience at the world level on open water race performances.
The discussion is well written. The authors explained the results obtained in their research by referring them to the literature.
However, I have a small comment relating to the "data analysis" part. The authors do not provide criteria for the interpretation of how they considered the top-10 finishing positions in each 10-km race were to be successful participants, but refer to their previous work (line 115). Considering that the work the authors refer to is published in a paid journal, I believe that such a description should be included in this part of the work.
Author Response
Response to Reviewer 3 Comments
First of all, we would like to thank the reviewers for their detailed revision of the manuscript and for their valuable suggestions that surely will improve the quality of the manuscript. Each of the reviewer comments have been addressed below and changes have been indicated on the manuscript with red colour. Additionally, a small correction was done in lines 41 and 213 about the APP results of previous studies
Point 1: The authors do not provide criteria for the interpretation of how they considered the top-10 finishing positions in each 10-km race were to be successful participants, but refer to their previous work (line 115). Considering that the work the authors refer to is published in a paid journal, I believe that such a description should be included in this part of the work
Response 1: The explanation as to why the top-10 finishing positions were considered as successful participants is given in the following sentence: “This metric was used as the top-10 positions in the WCH race the year prior to the OG are directly classified for the next OG edition”